# Sample-Aware Dual Actions for Prompt Optimization

## Abstract

In recent years, large language models (LLMs) have achieved remarkable progress in reasoning, question answering, and decision-making tasks in natural language processing. High-quality prompts play a crucial role in guiding LLMs to generate outputs that meet expectations. However, manually designing effective prompts for specific tasks is often time-consuming and heavily reliant on expertise, limiting the scalability and efficiency of model applications. Consequently, automated prompt optimization has become an important direction for enhancing LLM performance. To address this, we propose a sample-aware dual actions Monte Carlo Tree Search (MCTS) framework for automated prompt optimization, enabling the search process to leverage sample performance for more effective optimization. This method not only efficiently utilizes training samples to guide prompt improvement but also directs the optimization trajectory based on the overall state of the training samples. We validate our framework on the Big-Bench Hard (BBH) and MMLU datasets, and experimental results demonstrate that it outperforms traditional prompt optimization methods and recent baselines in both accuracy and optimization stability.

## 1 Instruction

Prompts serve as the critical interfaces that directly determine the output quality and task performance of large language models (LLMs), making prompt optimization an important research direction for enhancing LLM capabilities. As research has progressed, prompt optimization methods have gradually evolved from manual template design to automated learning and search approaches.

In early studies, prompts relied primarily on manual design. For example, few-shot prompting Brown et al. (2020) inserts a small number of examples into the prompt to help the model better align with task objectives; role-based prompting Kong et al. (2024) and Chain-of-Thought (CoT) strategies Kojima et al. (2022) guide the model to generate desired answers by enforcing templates through role assignments and reasoning chains. While these methods perform well on certain tasks, they heavily depend on human expertise and lack flexibility and transferability.

To reduce reliance on manual design, some researchers proposed continuous-representation-based prompt learning methods. Representative approaches include soft prompting Lester et al. (2021), prefix-tuning Li & Liang (2021) which optimize a trainable sequence of tokens in embedding space, transforming prompt learning into a parameter optimization problem. These methods can learn task-relevant prompts in an end-to-end manner and achieve strong performance on some tasks, but they often require task-specific parameter updates and have limited generalization ability.

Meanwhile, another line of work focuses on prompt optimization in discrete space. For instance, Local Prompt Optimization (LPO) Jain & Chowdhary (2025) performs fine-grained adjustments on key tokens; GAAPO Sécheresse et al. (2023) leverages genetic algorithms to search the global prompt space through mutation and selection mechanisms; PhaseEvo Cui et al. (2025) employs a multi-stage evolutionary strategy, optimizing different aspects of prompts in successive stages to gradually improve task performance.

Further, researchers began framing prompt optimization as a strategy search problem. For example, PromptAgent Wang et al. (2024) uses Monte Carlo Tree Search to model prompt construction as a series of reflection and improvement actions. The core advantage of PromptAgent lies in its sys-

tematic action planning via tree search, using failed samples for error reflection to improve prompts and avoid repeated mistakes. While these methods have achieved significant improvements, they often assume all task samples contribute equally, without explicitly modeling differences in sample informativeness or difficulty.

In reinforcement learning, some methods highlight the importance of sample-aware learning. Hindsight Experience Replay (HER)Andrychowicz et al. (2017) converts failed samples into useful learning signals through post-hoc reinterpretation, allowing failed experiences to be utilized; Self-Imitation Learning (SIL)Oh et al. (2018) emphasizes reusing successful samples by imitating historically high-reward trajectories to reinforce existing knowledge. These approaches demonstrate that dynamically attending to sample value and importance can significantly enhance learning efficiency.

Inspired by these insights, we propose an MCTS-based prompt optimization method that abstracts optimization actions into two core strategies: **inductive actions**, which leverage successful experiences to guide search, and **reflective actions**, which adapt strategies based on underperforming samples. To support this dual-action framework, we introduce a dynamic sample pool that continuously monitors sample quality, providing sample-sensitive guidance during the search while mapping the overall state into a global feedback signal to direct MCTS exploration. By integrating local sample awareness with global state feedback, our method achieves stable and efficient prompt optimization under dynamic sample distributions, while addressing the limitations of traditional search strategies that overlook sample heterogeneity.This establishes a principled framework for sample-aware prompt optimization.

## 2 METHODOLOGY

Our study presents an original framework for automated prompt optimization, built on the principles of sample-aware learning and Monte Carlo Tree Search. The central idea is to abstract the optimization process into two complementary actions—*Failure-Aware Reflection* and *Success-Aware Induction* where negative and positive samples provide corrective and generalizing signals for prompt optimization. These dual actions, together with a dynamic sample pool, enable efficient, adaptive, and stable prompt optimization across diverse tasks.

During the development of this framework, we observed that PromptAgent Wang et al. (2024) shares certain conceptual similarities, particularly in its use of Monte Carlo Tree Search and reflective analysis of failed cases. However, our approach is fundamentally original: unlike PromptAgent, which focuses solely on failure-aware reflection without explicitly summarizing successful patterns, our method formalizes a dual-action paradigm that combines both reflection and induction. Furthermore, PromptAgent treats all samples equally, ignoring differences in informational value. As a result, its effectiveness may decline on large datasets or when computational resources are constrained. Our framework introduces a mathematically grounded dynamic sample pool to quantify sample usefulness, enabling MCTS to prioritize high-value samples and efficiently explore the prompt space in a closed-loop feedback system.

### 2.1 MOTIVATION AND ACTION SPACE DESIGN

Following PromptAgent, we adopt Monte Carlo Tree Search (MCTS) as the backbone for prompt optimization, where each node corresponds to a prompt state and edges represent possible optimization actions. The search process relies critically on the design of the action space, since it defines the set of transformations that guide optimization trajectories.

A straightforward approach is to manually define fine-grained actions—such as rewriting instructions, adding examples, or adjusting role assignments. However, this manual design heavily depends on human prior knowledge: potentially useful actions may be overlooked, while low-value actions are often introduced. The former limits the exploration of high-efficiency strategies, and the latter wastes computational resources by diluting search signals and slowing convergence. Consequently, the effectiveness of the framework is tightly coupled with the quality and completeness of human prior specification.

A key takeaway from these limitations is that prompt optimization cannot rely on an exhaustive set of handcrafted operations. Excessively narrow action spaces may miss high-yield strategies,

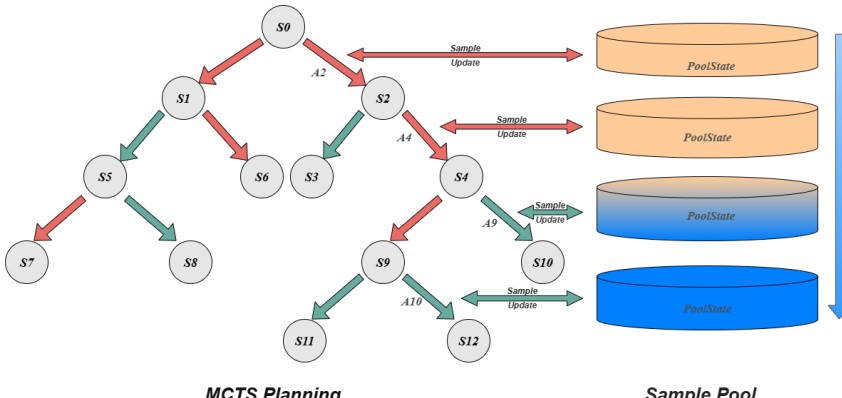

Figure 1: Overall framework. The left part shows MCTS where each node represents a prompt state and edges correspond to actions. The right part illustrates the dynamic sample pool, which is sampled and updated within each action, and whose evolving state in turn guides the tree search.

while overly broad ones dilute search signals, reducing efficiency. What is needed is a higher-level abstraction that captures the essential logic of optimization, rather than enumerating all possible surface-level edits.

## 2.2 Sample-Aware Dual Actions

Learning and reasoning processes in philosophy, logic, and cognitive science often rely on complementary Reflection–Induction strategies: errors are identified and corrected through reflection, while effective patterns are extracted and generalized through induction. When these strategies are iteratively applied in a feedback loop, they collectively form a closed-loop learning mechanism that continuously improves performance. Inspired by this, we define following actions.

**Failure-Aware Reflection:** For low-reward or erroneous samples, the system reflects on potential issues and adjusts prompt configurations to prevent similar mistakes. Failed samples often contain critical information about model weaknesses, providing negative guidance. By prioritizing these low-reward samples, the system rapidly identifies weak points and reduces unproductive search paths.

**Success-Aware Induction:** For high-reward samples, the system extracts efficient prompt configurations and generalizes them in subsequent iterations. Successful samples reflect effective patterns, offering positive guidance. By sampling high-reward cases, the system replicates and amplifies successful experiences, accelerating overall convergence.

The two complementary actions *Failure-Aware Reflection* and *Success-Aware Induction* encapsulate the essential logic of prompt optimization, forming a balanced mechanism to both correct errors and reinforce effective patterns. This dual-action perspective can be seen as a refinement of the action modeling in PromptAgent, while the original framework randomly samples prompts and collects failures for optimization, it does not differentiate among samples in terms of their utility. Not all samples contribute equally to improving prompt performance, and blindly treating them as equivalent may waste search budget. In the context of a limited optimization process, a key question emerges: how can we prioritize samples that provide the most informative guidance, thereby maximizing the efficiency of each iteration?

## 2.3 Dynamic Sample Pool

To effectively implement Failure-Aware Reflection and Success-Aware Induction, it is crucial to determine which samples should be emphasized during optimization. A naive strategy might directly select the lowest- or highest-reward samples—failures for reflection and successes for induction. However, reward alone is insufficient: some failures may be too trivial or noisy to provide meaningful guidance, while some successes may be redundant and offer little transferable insight.

Therefore, instead of relying solely on reward values, we design a dynamic sample pool that integrates additional factors such as difficulty, variance, and recent improvement. This ensures that the chosen samples are not only representative of success or failure, but also the most *informative* for driving efficient prompt optimization.

The sample pool $\mathcal{P} = \{s_1, s_2, \ldots, s_N\}$ is introduced to dynamically capture task-specific feedback and guide search. Unlike methods that treat all samples equally, we argue that *not all samples are equally informative*: some failures provide strong diagnostic signals for reflection, while certain successes provide valuable generalization patterns. Thus, the pool is designed to differentiate between easy, informative, and hard samples, and to modulate the optimization process accordingly.

### 2.3.1 SAMPLE METRIC

During prompt optimization, each sample $s_i$ maintains a record of its historical rewards:

$$\mathcal{R}_i = \{r_{i,1}, r_{i,2}, \ldots, r_{i,T_i}\}, \quad r_{i,t} \in [0, 1],$$

where $T_i$ is the number of evaluations, and $b_i$ is the baseline reward evaluated from original prompts.

**Reward Statistics.** We compute the empirical mean and variance to capture overall difficulty and stability:

$$\bar{r}_i = \frac{1}{T_i} \sum_{t=1}^{T_i} r_{i,t}, \quad \sigma_i^2 = \frac{1}{T_i - 1} \sum_{t=1}^{T_i} (r_{i,t} - \bar{r}_i)^2$$

Intuitively, a low mean indicates persistent difficulty, while a high variance suggests instability and potential room for improvement.

**Informative Score.** To prioritize samples for prompt optimization, we define an *informative score* $I_i$ that combines three complementary factors with adjustable weights:

$$I_i = \alpha \, D_i + \beta \, G_i + \gamma \, V_i,$$

where $\alpha, \beta, \gamma \geq 0$ and $\alpha + \beta + \gamma = 1$.

The three components are defined as:

$$D_i = 4b_i(1 - b_i), \quad G_i = \frac{\text{mean}(\text{recent } r_{i,t}) - b_i}{1 + \epsilon}, \quad V_i = \frac{\sigma_i^2}{\sigma_{\max}^2 + \epsilon}.$$

**Difficulty** $D_i$: Acts as a prior regularizer to prevent overemphasis on extreme samples. Very easy or very hard samples provide unstable optimization signals, while samples with moderate difficulty offer more balanced feedback. Unlike variance, which captures empirical differentiation across candidates, difficulty serves as a prior estimate that stabilizes the contribution of each sample.

**Recent Gains** $G_i$: Measures the improvement potential based on recent reward trends. Samples where the model can still make progress are prioritized.

**Variance** $V_i$: Reflects uncertainty in the model's predictions. High-variance samples are more informative for learning, guiding the model to areas with inconsistent performance.

By combining these factors, $I_i$ highlights samples that are challenging yet learnable and uncertain, thus maximizing the expected benefit for prompt optimization.

### 2.3.2 SAMPLE PRIORITY

To effectively leverage both failures and successes in prompt optimization, we assign each sample $i$ a priority score that guides reflection and induction. This priority combines information about the sample's reward, informativeness, and prediction uncertainty, allowing the optimization process to focus on the most informative and actionable examples.

**Sample-aware priority:** We define the priority of each sample $i$ for reflection and induction as

$$\text{priority}_i = \begin{cases} (1 - \bar{r}_i) + \theta I_i + \lambda \min\left(1, \frac{\sigma_i^2}{\sigma_{\text{unstable}}^2}\right), & \text{for reflection (negative-aware)} \\ (\bar{r}_i - b_i) + \mu I_i - \delta \min\left(1, \frac{\sigma_i^2}{\sigma_{\text{unstable}}^2}\right), & \text{for induction (positive-aware)} \end{cases}$$

where $\bar{r}_i$ is the recent reward, $I_i$ is the informative score, and $\sigma_i^2$ reflects prediction uncertainty. Failures with high difficulty or high variance are prioritized for reflection, as they highlight weaknesses of the current prompt and provide strong corrective signals. Successful samples with stable improvements are emphasized for induction, since they reveal reusable patterns that can be generalized to guide subsequent prompt optimization.

### 2.3.3 SAMPLE POOL QUALITY METRIC

The quality and composition of the sample pool critically affect the efficiency of prompt optimization. A well-characterized pool allows the system to identify which samples provide the most informative guidance, which are already mastered, and which remain challenging. To formalize this, we define a *pool quality metric* that evaluates the state of the sample pool, capturing both convergence and remaining learning potential. This metric serves as a global signal for guiding exploration and prioritizing actions in the MCTS process.

To obtain a conservative estimate of individual sample quality, we adopt the *lower bound of the Wilson score interval*. Given an empirical accuracy estimate

$$\hat{p}_i = \frac{\text{correct}_i}{T_i},$$

the Wilson interval provides a statistically robust confidence interval for the true proportion, particularly under small sample sizes where normal approximations are unreliable. Formally, the lower confidence bound is defined as

$$\text{LB}_i = \frac{\hat{p}_i + z^2/(2T_i) - z\sqrt{\hat{p}_i(1 - \hat{p}_i)/T_i + z^2/(4T_i^2)}}{1 + z^2/T_i},$$

where $T_i$ is the number of trials and $z$ is the quantile of the standard normal distribution corresponding to the desired confidence level (e.g., $z = 1.96$ for 95%). Using the lower bound rather than the empirical mean ensures a *statistically conservative estimate*, reducing the risk of overestimating uncertain samples and yielding a more stable measure for guiding MCTS.

### 2.4 SAMPLE-POOL-GUIDED MCTS INTEGRATION

Our integration of the sample pool into MCTS operates along two complementary dimensions.

**Action Selection Adjustment.** During tree expansion, the selection probability of each candidate action $a_i$ is determined by both its empirical statistics and the current state of the sample pool. Each action maintains two counters: the total execution count $u_i$ and the failure count $f_i$, which records unproductive applications that did not improve outcomes. For induction actions targeting high-reward samples, a failure occurs if all tested instances produce incorrect outputs; for reflection actions targeting low-reward samples, a failure occurs if all tested instances are already correct and the action provides no corrective effect.

The final action logit combines these empirical statistics with a pool-aware bias $b_i$:

$$\ell_i = -(f_i + u_i) + b_i, \quad b_i = \begin{cases} +H(c_{\text{pool}}) \cdot \tanh(\text{easy\_ratio} - \text{hard\_ratio}) \cdot s, & a_i \in \text{Inductions}, \\ -H(c_{\text{pool}}) \cdot \tanh(\text{easy\_ratio} - \text{hard\_ratio}) \cdot s, & a_i \in \text{Reflections}. \end{cases}$$

Intuitively, the bias $b_i$ adjusts the preference between induction and reflection actions based on the sample pool composition. When the pool contains mostly easy samples, induction actions are favored while reflection actions are suppressed. Conversely, when hard samples dominate, reflection actions are promoted and induction actions are moderated. By combining empirical statistics with the pool-aware bias, the system adaptively prioritizes actions that are most likely to be productive given the current distribution of samples.

**Exploration–Exploitation Balance.** In addition to action-specific biases, the overall exploration–exploitation trade-off is dynamically modulated by the composition of the sample pool. We categorize samples into three types: *easy* (already mastered), *informative* (high potential to guide optimization), and *hard* (challenging or underexplored). Intuitively, when the pool is dominated by easy samples, most patterns have been captured, so exploration can be reduced and exploitation

Table 1: Performance on BBH tasks. Our framework achieves the best overall average (0.901), surpassing PromptAgent (0.862). Gains are largest on complex reasoning tasks, including *Epistemic* (+6.5%) and *Object Counting* (+8.1%), demonstrating the benefit of dynamic sample guidance and dual-action optimization. Simpler tasks like *Penguins Table* are near saturation, where all methods perform comparably.

| Method | Causal Judgement. | Epistemic. | Geometric Shapes. | Object Counting. | Penguins Table. | AVG. |
|---|---|---|---|---|---|---|
| HumanZS | 0.53 | 0.65 | 0.42 | 0.596 | 0.556 | 0.55 |
| HumanFS | 0.57 | 0.74 | 0.45 | 0.654 | 0.683 | 0.619 |
| HumanFS-Analytic | 0.49 | 0.78 | 0.45 | 0.472 | 0.708 | 0.581 |
| CoTZS | 0.57 | 0.836 | 0.895 | 0.916 | 0.924 | 0.828 |
| CoTFS | 0.65 | 0.818 | 0.865 | 0.904 | 0.924 | 0.832 |
| CoTFS-Analytic | 0.62 | 0.858 | 0.845 | 0.866 | 0.848 | 0.808 |
| PromptAgent | 0.67 | 0.896 | 0.86 | 0.912 | 0.974 | 0.862 |
| Our | **0.68** | **0.954** | **0.91** | **0.986** | **0.974** | **0.901** |

emphasized. Conversely, when the pool contains many informative or hard samples, exploration is encouraged to discover new strategies and extract useful information.

To formalize this intuition, we define an adaptive exploration weight:

$$w_{\text{explore}} = w_{\min} + (w_{\max} - w_{\min}) \left(\text{informative\_ratio} + \text{hard\_ratio}\right)^{\kappa},$$

where $w_{\min}$ and $w_{\max}$ set the allowable exploration range, and $\kappa > 0$ controls sensitivity to pool composition. Here, *informative ratio + hard ratio* represents the fraction of samples that suggest remaining learning potential, naturally increasing exploration when many samples are underexploited. In practice, we set the exploration bounds to empirical values that balance sufficient search with stability. A reasonable choice is $w_{\min} = 0.2$ and $w_{\max} = 2.0$, which ensures that even when the pool is dominated by easy samples, a small amount of exploration is preserved, while the presence of many informative or hard samples triggers significantly increased exploration. These values can be further tuned according to the depth of the MCTS tree, the size of the action space, and the computational cost of evaluating individual actions.

The UCT node selection score is computed as

$$\text{Score}(a) = Q(a) + w_{\text{explore}} \cdot \sqrt{\frac{\ln N_{\text{parent}}}{1 + N(a)}}, \quad a^* = \arg\max_a \text{Score}(a),$$

where $Q(a) = \frac{1}{N(a)} \sum_{i=1}^{N(a)} R_i$ is the average reward collected from rollouts passing through action $a$, representing the empirical value of the action. $N_{\text{parent}}$ and $N(a)$ denote the visit counts of the parent node and the child node corresponding to action $a$, respectively. The exploration weight $w_{\text{explore}}$ replaces the conventional constant term, allowing the search intensity to adapt dynamically according to the current composition of the sample pool.

Every executed action, including both expansion and rollout, updates the sample pool, which in turn recalibrates convergence scores, variances, and informative ratios. This closed-loop feedback mechanism ensures adaptive behavior: failures steer the system toward reflection actions to correct remaining errors, successes promote induction actions to generalize successful patterns, and the evolving pool composition dynamically regulates both action selection and the strength of exploration through the adaptive $w_{\text{explore}}$. Finally, following the strategy used in PromptAgent, the framework selects the optimal prompt by tracing the path with the highest average reward in the MCTS tree and choosing the node along this path with the maximum reward, ensuring that the selected prompt represents the most effective combination of actions discovered during search.

## 3 EXPERIMENTS

### 3.1 EXPERIMENTAL SETUP

**Datasets.** We evaluate our framework on a diverse set of tasks to demonstrate its effectiveness in prompt optimization. First, following the setup of PromptAgent, we include five reasoning tasks from *BBH (Big-Bench Hard)* Suzgun et al. (2023), such as *Casual Judgement*, *Object Counting*, and *Geometric Shapes*, covering logical inference, quantitative reasoning, and structural understanding.

Table 2: Performance on domain-specific tasks. Our framework achieves the highest average (0.725), outperforming PromptAgent (0.681). Notable gains are observed in challenging domains: *Multi label Ethos* (+11.5%), *MedQA* (+3.2%), and *CaseHold* (+3.5%), highlighting the advantage of dynamic prompt adaptation even for factual and expert knowledge.

| Method | Pro Engineering. | Pro Business. | Multi label Ethos. | MedQA. | CaseHold. | AVG |
|---|---|---|---|---|---|---|
| HumanZS | 0.485 | 0.47 | 0.16 | 0.61 | 0.64 | 0.473 |
| HumanFS | 0.51 | 0.47 | 0.225 | 0.685 | 0.67 | 0.512 |
| HumanFS-Analytic | 0.46 | 0.71 | 0.275 | 0.755 | 0.655 | 0.571 |
| CoTZS | 0.72 | 0.795 | 0.185 | 0.765 | 0.67 | 0.627 |
| CoTFS | 0.725 | 0.86 | 0.295 | 0.795 | 0.71 | 0.677 |
| CoTFS-Analytic | 0.745 | 0.855 | 0.395 | 0.815 | 0.71 | 0.704 |
| PromptAgent | **0.755** | 0.815 | 0.32 | 0.795 | 0.72 | 0.681 |
| Our | 0.75 | **0.875** | **0.435** | **0.82** | **0.745** | **0.725** |

Second, we incorporate five domain-specific tasks from professional benchmarks: *Pro Engineering* (engineering) and *Business* (business) subsets from *MMLU* Hendrycks et al. (2021), *MedQA* (medicine) Jin et al. (2020), *CaseHold* (law) Zheng et al. (2021), and *ETHOS* (ethics) Mollas et al. (2022). These tasks evaluate prompt generalization in specialized contexts requiring precise reasoning and domain knowledge. Finally, we consider a more challenging set, *BBEH (Big-Bench Extra Hard)* Kazemi et al. (2025), which introduces higher complexity and stricter consistency requirements. We also experimented with *GSM8K* Cobbe et al. (2021), *MultiArith* Roy & Roth (2015), and the *Temporal Sequences* subset from BBH, but these are omitted from the main presentation due to near-saturated performance across all methods. Overall, these datasets provide a comprehensive evaluation spanning general reasoning, domain-specific knowledge, and challenging reasoning scenarios.

**Baselines.** We compare our method with three categories of baselines: human-designed prompts, Chain-of-Thought (CoT) prompts, and automated prompt optimization methods. Human prompts are manually written instructions that reflect common practices in prompt engineering and often originate from the datasets themselves. In addition to the plain zero-shot version, we also consider a few-shotBrown et al. (2020) variant that supplements the prompt with task-specific exemplars, as well as an extended few-shotWei et al. (2023) variant where the exemplars include not only input–output pairs but also worked-out reasoning steps. Chain-of-Thought promptsKojima et al. (2022) build directly on these human-designed variants by explicitly instructing the model to "think step by step," thereby encouraging intermediate reasoning before producing the final answer. This allows us to examine whether improvements come from automated optimization or simply from exposing the model to structured reasoning. Finally, we benchmark against PromptAgentWang et al. (2024), a recent automated optimization method that frames prompt planning as a Monte Carlo Tree Search over edit actions: it performs trial-and-error rollouts, collects failure cases and reflection signals, and uses these signals to iteratively refine prompts. Conceptually, PromptAgent can be regarded as an ablated (simplified) variant of our framework approach, since it lacks the combined structural search and semantic action-planning mechanisms present in our full system.

**Experimental Details.** We employ two categories of models in our experiments: an optimization model and a base evaluation model. The optimization model, *DeepSeek-R1* Guo et al. (2025), is responsible for executing optimization actions in both PromptAgent and our framework, performing error analysis and pattern induction, and generating analytic variants of few-shot prompts. The base model, *Qwen-Flash* Bai et al. (2023), is used to assess training samples during optimization, compute rewards from validation data in MCTS rollouts, and evaluate the final optimized prompts on test sets.

For each single task in BBH and domain-specific datasets, we perform three independent runs. Due to the intrinsic stochasticity of the optimization model (sampled at temperature = 1), we fix the random seed to 42 during sample processing, ensuring controlled and reproducible experiments. The base evaluation model, in contrast, is used deterministically with temperature = 0, to ensure stable and consistent assessment of samples and prompt performance. On the supplementary BBEH dataset, only a single run is reported because the high frequency of model invocations incurs extremely high API costs; nevertheless, our method consistently outperforms baselines across tasks, demonstrating effectiveness under limited resources.

Table 3: On the harder BBH tasks (BBEH), although only a single experimental run was conducted due to computational constraints—so formal statistical significance cannot be assessed—our framework still demonstrates clear advantages over baselines. The gains on *Object Counting* (+60%) and *Boolean Expressions* (+13.5%) indicate that the dual-action optimization and sample-guided search are effective even in highly challenging scenarios.

| Method | Casual Judge. | Temporal Sequences. | Object Counting. | Boolean Expressions. | AVG |
|---|---|---|---|---|---|
| HumanZS | 0.425 | 0.0125 | 0 | 0.3375 | 0.194 |
| HumanFS | 0.4625 | 0.0125 | 0 | 0.2375 | 0.178 |
| HumanFS-Analytic | **0.5875** | 0.0375 | 0.0375 | 0.175 | 0.209 |
| CoTZS | 0.525 | **0.2625** | 0.175 | 0.475 | 0.359 |
| CoTFS | 0.575 | 0.2375 | 0.2125 | 0.375 | 0.350 |
| CoTFS-Analytic | 0.55 | 0.1625 | 0.175 | 0.3625 | 0.312 |
| PromptAgent | 0.5375 | 0.2 | 0.1875 | 0.3875 | 0.328 |
| Our | 0.4625 | 0.225 | **0.3** | **0.525** | **0.378** |

All main experiments employ a standard MCTS configuration, with 10 rollouts per node, a tree width of 3, and a maximum tree depth of 5. To control computational cost, we utilize common MCTS early-stopping mechanisms, terminating rollouts once further exploration is unlikely to yield significant improvement. This configuration balances efficiency and exploration, ensuring the search process remains tractable while maintaining high-quality prompt optimization.

Finally, in the ETHOS experiments, not all samples can be processed successfully by the base model, which occasionally rejects inputs as invalid. During optimization and evaluation, such samples are automatically skipped. Since the same filtering mechanism applies uniformly across all compared methods, this does not introduce bias and ensures fair comparison.

## 3.2 EXPERIMENTAL RESULTS AND ANALYSIS

**Comparison with Baselines.** Across both BBH reasoning and domain-specific tasks, our framework consistently outperforms all baselines, demonstrating the effectiveness of dual-action optimization with dynamic sample guidance. On BBH tasks, the largest gains appear in complex settings such as *Epistemic* (+6.5%) and *Object Counting* (+8.1%), which require multi-step logical reasoning and quantitative computation. Simpler tasks, such as *Penguins Table*, already approach ceiling performance, leaving limited room for improvement. On domain-specific tasks, our method achieves strong improvements, particularly on *Multi label Ethos* (+11.5%), *MedQA* (+3.2%), and *CaseHold* (+3.5%), indicating that dynamic adaptation remains beneficial even for factual and expert knowledge. On the harder BBH subset (BBEH), our framework improves the overall average by +15.2% over PromptAgent, with substantial gains on *Object Counting* (+60%) and *Boolean Expressions* (+13.5%). Due to the high computational cost, BBEH results are based on a single run; thus, statistical significance cannot be reliably estimated for these tasks, and some task-specific scores, e.g., *Casual Judge*, may fluctuate due to stochasticity. These results confirm the robustness and general effectiveness of our framework across BBH and domain-specific benchmarks, with consistent improvements observed across multiple tasks.

**Ablation Study.** We evaluate three configurations to quantify the contribution of each component: *Beam With Pool* (beam search guided by the sample pool), *MCTS Only* (dual-action MCTS without sample guidance), and the *Full Framework* (combining both mechanisms). On BBH and domain-specific tasks, both Beam With

Table 4: Ablation study of framework configurations on BBH, harder BBH, and domain-specific tasks, showing the complementary benefits of sample-guided search and MCTS in improving overall performance.

| Task Category | Beam With Pool. | MCTS Without Pool. | Full. |
|---|---|---|---|
| Big Bench Hard | 0.783 | 0.841 | 0.901 |
| Domain-Specific | 0.662 | 0.677 | 0.725 |
| Big Bench Extra Hard | 0.382 | 0.402 | 0.378 |

Pool and MCTS Only outperform baseline methods, highlighting complementary advantages: sample-guided search accelerates convergence by focusing on informative examples, while MCTS enhances exploration of the prompt space and captures complex dependencies. Integrating these components in the Full configuration yields the highest overall performance, especially on challenging reasoning tasks such as *Object Counting* and *Epistemic*, demonstrating the synergy between

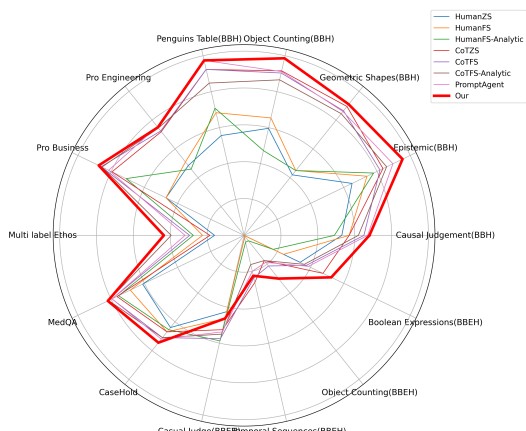

Figure 2: Comparison with Baselines

structured search and dynamic sample guidance. On the harder BBEH tasks, the Full setup achieves competitive performance (0.378), slightly below MCTS Only (0.402) in this single-run evaluation, reflecting stochastic variability; nevertheless, it confirms the robustness and general effectiveness of the combined approach across the majority of benchmarks.

In summary, our experiments show that the proposed framework consistently improves performance on general reasoning, domain-specific, and challenging tasks. Ablation studies confirm that its components provide complementary benefits and are jointly necessary for maximum gains. Overall, these results validate that dual-action prompt optimization with dynamic sample guidance is an effective and general approach for enhancing task performance. Figure 2 illustrates the relative improvements of our method over key baselines across multiple task categories.

## 4 CONCLUSION

In this work, we proposed a novel framework for structured prompt optimization that is both intuitive and grounded in human-like optimization behavior. Our approach abstracts the essential aspects of human prompt refinement by modeling dual core actions—reflection and induction—while explicitly considering interactions with the available learning material. By doing so, the framework aims to maximize the informational utility of the sample pool under limited computational resources and time constraints, effectively guiding the search for high-quality prompts.

Empirical results on both general reasoning and domain-specific benchmarks demonstrate that our method consistently outperforms existing baselines, highlighting the effectiveness of integrating action planning with sample-aware guidance. The framework provides a systematic, interpretable approach to prompt optimization, offering a principled alternative to purely heuristic or manual strategies.

Despite these advantages, our method still has limitations. Heuristic search in the prompt optimization domain remains inherently costly due to frequent model calls, and the design of the sample pool, while intuitive, could be further improved. Specifically, more sophisticated strategies for evaluating samples and maintaining pool statistics may yield additional gains in efficiency and robustness. We emphasize that our primary contribution lies in the framework itself; specific implementation details can likely be further optimized, suggesting avenues for future research.

Overall, this work establishes a structured, human-inspired approach to prompt optimization that is general, adaptable, and interpretable, while also identifying key challenges and opportunities for continued improvement in efficient prompt engineering.

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

# A  APPENDIX

## A.1  PARAMETER SENSITIVITY

We further examine the robustness of our framework with respect to key parameters in the sample pool design. All experiments in this section are conducted on the *Boolean Expressions* task, with the number of MCTS rollouts per node set to 5 to control computational cost, while other MCTS parameters remain unchanged.

Specifically, we focus on the weighting parameters in reflection-action sampling priorities: the informativeness weight $\lambda$ and the variance weight $\theta$. The sample pool maintains past action execution results and sample states to guide reflection-action selection, where $\lambda$ controls the preference for informative samples and $\theta$ controls the preference for high-variance samples. As shown in Figure 3, we systematically vary $\lambda$ and $\theta$ and observe the resulting accuracy. Several patterns emerge: When $\lambda = 0.4$, accuracy slightly decreases as $\theta$ increases from 0.16 to 0.24 ($0.475 \rightarrow 0.425$), indicating that too low or too high a variance weight may slightly affect performance. When $\lambda = 0.5$, overall accuracy reaches its maximum, with the highest value 0.55 at $\theta = 0.16$, suggesting that a moderate variance weight achieves a good balance, allowing reflection actions to better exploit informative high-variance samples. When $\lambda = 0.6$, accuracy again decreases to the range of 0.425–0.475, showing that overly high variance weighting may bias selection toward overly volatile samples, slightly reducing optimization stability.

Overall, accuracy fluctuates moderately across different parameter combinations (0.375–0.55), demonstrating that the framework is robust with respect to sample pool parameters. The results also indicate that a combination of moderate variance weight $\theta$ and moderate informativeness weight $\lambda$ slightly improves reflection-action sampling and prompt optimization performance. The heatmap provides a visualization of this interaction and offers intuitive guidance for selecting sample pool parameters.

## A.2  ACTION GUIDANCE PROMPTS

In our framework, both `FailureAwareAction` and `SuccessAwareAction` rely on structured guiding prompts to steer the language model's reasoning, analysis, and rewriting behavior. These prompts are explicitly designed to provide contextual information about the current prompt, examples from the sample pool, and evolution history of prompts. The design of these prompts is inspired by the `PromptAgent` methodology, ensuring systematic and interpretable interactions between actions and model behavior.

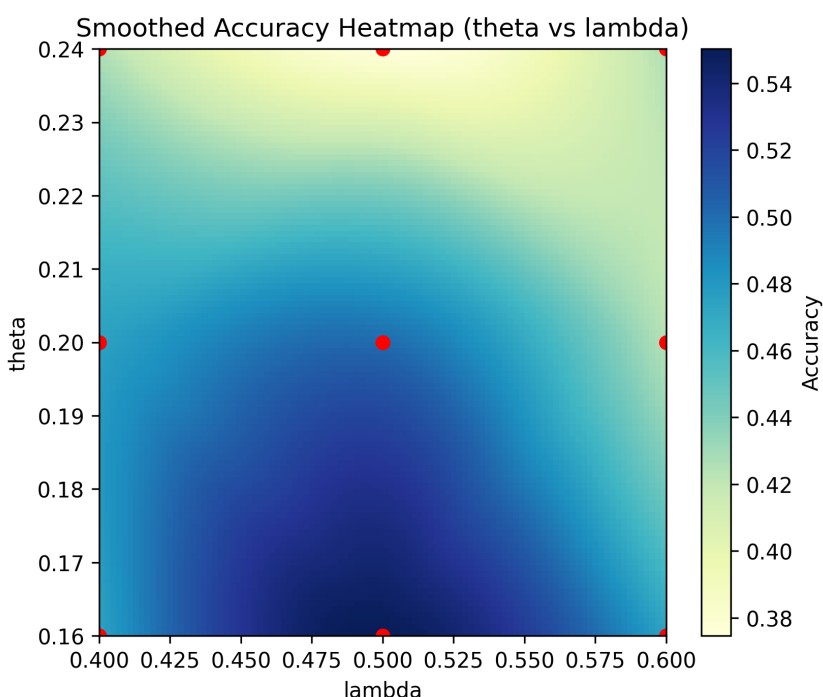

Figure 3: Parameter Sensitivity.

### A.2.1  FAILURE-AWARE ACTION PROMPTS

The `FailureAwareAction` uses two main prompts: one for analyzing failure cases and one for rewriting the prompt.

**Failure Analysis Prompt:**

```
I'm writing prompts for a language model designed for a task.

### Information about the current prompt: ###
My current prompt: {current_prompt}

This prompt leads to incorrect responses for the following examples:
    {wrong_examples}

### Analysis of Failure Cases: ###
For each wrong example, you should carefully analyze why model response
    wrong answer, why my prompt leads to wrong answer. Provide
    comprehensive analysis of the common failure modes, pitfalls, or
    ambiguities in the prompt that may have contributed to these errors.
    List me all suggest improvements to the prompt to ensure better
    generalization.
```

**Failure-Aware Rewriting Prompt:**

```
I'm optimizing a prompt for a language model on a specific task.

### Information about the current prompt: ###
My current prompt: {current_prompt}

This prompt leads to incorrect responses for the following examples:
    {wrong_examples}

Some analysis and suggestions for avoid wrong answers: {analysis}
```

```
There are a list of former prompts evolve to the current prompt, and
    each prompt is modified from its former prompts:
    {trajectory_prompts_str}

### Requirements for the new prompt: ###
The new prompt should solve the current prompt's problems. The new
    prompt should consider the list of prompts and evolve based on the
    current prompt. Please rewrite the prompt accordingly. Only output
    the new prompt.

### Suggestions: ###
1. You may consider adding similar failure examples as 'few-shot'
    references to help the model avoid repeating similar mistakes.
2. You could reinforce reasoning steps that were missed in previous
    iterations, such as [specific reasoning step].
3. There might be value in avoiding common mistakes such as [specific
    mistake, e.g., ambiguity in instructions].
4. Providing additional context or clarifications may help the model
    better understand edge cases and task-specific details.
5. You could focus on guiding the model step by step to improve its
    generalization ability in future tasks.
```

### A.2.2  SUCCESS-AWARE ACTION PROMPTS

The `SuccessAwareAction` also uses two main prompts: one for analyzing successful examples and one for rewriting the prompt.

**Success Analysis Prompt:**

```
I'm optimizing a prompt for a language model on a specific task.

### Information about the current prompt: ###
My current prompt: {current_prompt}

Here are some successful examples where the model's prediction matches
    the correct answer: {success_examples}

### Analysis of Successful Cases: ###
For each example, analyze why the model succeeded. Summarize the key
    reasoning strategies, invariants, decision rules, or intermediate
    steps that should be reinforced in the prompt to generalize better.
```

**Success-Aware Rewriting Prompt:**

```
I'm optimizing a prompt for a language model.

### Information about the current prompt: ###
My current prompt: {current_prompt}

Here are some successful examples where the model's prediction matches
    the correct answer: {success_examples}

The following strengths and reasoning strategies were identified from
    successful examples: {analysis}

There are a list of former prompts evolve to the current prompt, and
    each prompt is modified from its former prompts:
    {trajectory_prompts_str}

### Requirements for the new prompt: ###
The new prompt should consider the list of prompts and evolve based on
    the current prompt. Please rewrite the prompt to incorporate and
    emphasize these strengths while keeping it concise and clear. Only
    output the new prompt.
```

```
### Suggestions: ###
1. You might want to reinforce successful reasoning steps that
    contributed to the correct answers (e.g., [specific reasoning step]).
2. You could add 'few-shot' examples based on successful cases to guide
    the model in future tasks, especially for [specific type of task].
3. It might be helpful to keep the prompt concise while emphasizing the
    key reasoning strategies that worked in these examples.
4. You may want to ensure that the model can generalize these strengths
    to new tasks or edge cases.
5. You could highlight decision rules or key steps that contributed to
    success, and ensure they are emphasized in the new prompt.
```

### A.3  DATASET SPLITS FOR PROMPT OPTIMIZATION

We organize our datasets into three categories: BBH tasks, domain-specific tasks, and BBEH tasks. For each category, we provide the number of samples in the training, validation, and test splits.

The training set serves as the source for prompt optimization and is stored in the sample pool. The validation set is used during training for rollouts to compute rewards and update tree node $Q$ values in the MCTS search. The test set is reserved for evaluating the performance of the optimized prompts.

Table 5: Dataset splits used for prompt optimization.

| Task Category | Train | Validation | Test |
|---|---|---|---|
| **BBH** | | | |
| Causal Judgement | 70 | 20 | 100 |
| Epistemic | 400 | 100 | 500 |
| Geometric Shapes | 120 | 30 | 200 |
| Object Counting | 240 | 60 | 500 |
| Penguins Table | 55 | 15 | 79 |
| **Domain-specific Tasks** | | | |
| MMLU Pro Engineering | 80 | 20 | 200 |
| MMLU Pro Business | 80 | 20 | 200 |
| ETHOS Multi-label | 80 | 20 | 200 |
| MedQA | 80 | 20 | 200 |
| CaseHold | 80 | 20 | 200 |
| **BBEH** | | | |
| Casual Judgement | 100 | 20 | 80 |
| Temporal Sequences | 100 | 20 | 80 |
| Object Counting | 100 | 20 | 80 |
| Boolean Expressions | 100 | 20 | 80 |

### A.4  CASE STUDIES OF OPTIMIZED PROMPTS

In this subsection, we present detailed case studies of two representative tasks from our benchmark: *Ethos Multi-label Hate Speech Classification* and *Geometric Shapes*. For each task, we show the original prompt, our optimized prompt, and provide an analysis of the improvements.

#### A.4.1  ETHOS MULTI-LABEL HATE SPEECH CLASSIFICATION

**Original Prompt:**

```
Classify text into multiple hate speech labels.
```

**Optimized Prompt:**

```
[basicstyle=\ttfamily\footnotesize, breaklines=true]
You are an expert AI for hate speech classification. Analyze the input
    text and classify it according to the specific definitions below.
    Your analysis must be precise and consider both explicit and
    implicit meaning.

### CRITICAL DEFINITIONS ###
```

```
* `violence`: 1 if the text explicitly threatens, glorifies, or calls
    for physical harm, injury, or death (e.g., "shoot," "hang," "kill,"
    "beat them up"). Calls for non-violent but harmful actions (e.g.,
    "deport," "fire them," "ban them") are NOT violence. Otherwise 0.
* `directed_vs_generalized`:
    * **1 (Directed)**: The text targets a specific, named individual or
        a specific individual referred to by a pronoun (e.g., 'you',
        'she', 'he', 'they' for one person) or a very small, identifiable
        group (e.g., "my neighbor Ahmed," "the employees of that store").
        If a text attacks a broad group but uses a singular pronoun as an
        example, it is still Generalized.
    * **0 (Generalized)**: The text targets a broad, abstract group based
        on a protected characteristic (e.g., "all Christians," "women
        are," "every Muslim").
* `gender`: 1 if the hate is primarily and explicitly based on gender
    identity or expression (e.g., misogyny, misandry, transphobia). The
    use of a gendered swear word alone, without context attacking the
    person's gender, is NOT sufficient. Otherwise 0.
* `race`: 1 if the hate is based on race or ethnicity. Otherwise 0.
* `national_origin`: 1 if the hate is based on country of origin,
    nationality, or immigrant status. Otherwise 0.
* `disability`: 1 if the hate is based on physical or mental disability.
    Otherwise 0.
* `religion`: 1 if the hate is based on religious affiliation, beliefs,
    or lack thereof. Otherwise 0.
* `sexual_orientation`: 1 if the hate is based on real or perceived
    sexual orientation (e.g., gay, lesbian, bisexual, asexual). This
    includes coded language, slurs, and stereotypes. Otherwise 0.

### IMPORTANT INSTRUCTIONS ###
1. A single text can contain multiple types of hate. Check all
    applicable labels.
2. Analyze the entire context. Pay special attention to coded language,
    metaphors, and implications that are tied to a protected
    characteristic.
3. CRITICAL: Determine the primary target and basis of the hatred. A
    text describing a person ("the gay man") is not hateful unless it
    attacks them for that characteristic. A text that uses one
    characteristic (e.g., race) to attack a group based on another
    characteristic (e.g., religion) should be labeled for the
    characteristic that is the target of the attack.
4. Distinguish between general pejoratives and hate based on protected
    characteristics. Words like "demon," "stupid," or "animal" are not
    inherently hate speech unless they are clearly linked to a protected
    class.
5. **Slur Context Analysis:** When encountering slurs, determine if they
    target a protected characteristic (e.g., racial slurs, homophobic
    slurs) or are used as general profanity. Slurs targeting protected
    groups should trigger the corresponding category.
6. **Term-to-Category Mapping:** Link specific terms to protected
    categories: LGBTQ+ terms -> sexual_orientation, disability terms ->
    disability, racial/ethnic terms -> race, religious terms ->
    religion, nationality/immigration terms -> national_origin,
    gender-specific attacks -> gender.
7. **Expression vs. Threat Distinction:** Expressions of hatred or
    prejudice alone do not constitute violence. Violence requires
    explicit threats, glorification, or calls for physical harm.
8. Be conservative. Only mark a category as 1 if there is clear and
    unambiguous evidence per the definitions.
9. Output MUST be a valid JSON object with the exact keys above and
    integer values (1 or 0).

### EXAMPLES TO GUIDE REASONING ###
... [examples omitted for brevity in code block] ...
```

```
### TASK ###
Analyze the following input text. Apply the definitions and reasoning
    shown in the examples.

Input: {insert_user_input_here}
Output:
```

The optimized prompt has several key advantages. First, it substantially increases the level of specificity and precision by defining each hate category in detail, distinguishing between violence and general expressions of hatred, and clarifying directed versus generalized attacks. Second, it instructs the model to analyze both explicit and implicit meanings, including slurs, metaphors, and contextual cues, which enhances multi-label classification robustness. Third, by including clear examples and step-by-step reasoning instructions, the prompt guides the model to consistently output structured JSON, reducing ambiguity and improving reproducibility. Overall, this optimized prompt achieves an approximate accuracy of **50%** on the ethos dataset, demonstrating a strong improvement over the original minimal prompt. The performance reflects the prompt's ability to capture nuanced hate speech characteristics across multiple protected attributes, making it an exemplary result of our structured prompt optimization framework.

### A.4.2 BIG BENCH HARD: GEOMETRIC SHAPES

The *Geometric Shapes* task requires identifying geometric figures from SVG path elements. We compare the original prompt with our optimized version below.

**Original Prompt:**

```
Name geometric shapes from their SVG paths
```

**Optimized Prompt:**

```
Name the geometric shape drawn by an SVG path element. Choose from the
    provided options.

**Instructions:**

1. **Preprocess the Path:**
   - Before parsing, check for redundant 'M' commands. If an 'M' or 'm'
      command is followed by coordinates that are **identical to the
      current point**, treat it as a continuation of the previous path
      segment, not a new subpath. This is a common technique to ensure
      precision.

2. **Parse the Path Commands:**
   - 'M'/'m' (MoveTo): Lifts the pen and moves to a new location. **Only
      starts a new subpath if it moves to a location *different* from
      the current point.**
   - 'L'/'l' (LineTo): Draws a straight line from the current point to a
      new point.
   - 'Z'/'z' (ClosePath): Draws a straight line back to the first point
      of the *current continuous path*.

3. **To identify the shape, follow these steps:**
   a. Break the path into separate **continuous paths**. A new
      continuous path begins only after an 'M'/'m' that moves to a new,
      distinct location.
   b. For a continuous path to be a polygon, it must be a closed shape.
      It is closed either by a final 'Z' command or by the last point
      being exactly equal to the very first point of that continuous
      path.
   c. **Crucially, count only the unique vertices in the continuous
      path.** The starting point (after an 'M') and all points from 'L'
      commands are vertices. The final point that closes the shape (via
      'Z' or an 'L' back to the start) is a duplicate and should
      **not** be counted again.
```

```
    d. A single '<path>' can contain multiple shapes. The overall shape
       is defined by the largest closed continuous path. If no
       continuous path is closed, the answer may be "line".

4. **Avoid common mistakes:**
   - Do not simply count all coordinate pairs. You must interpret the
     commands.
   - An 'M' that moves to a new location is the start of a new path and
     is not a vertex of the previous path.

**Examples of Common Errors and Corrections:**

* **Error:** Treating all 'M' commands as absolute breaks, missing a
   single continuous polygon.
   * Path: 'M A L B M B L C L D L A'
   * **Old Incorrect Analysis:** Two subpaths (line AB and open path
     B->C->D->A). Answer: "line".
   * **New Correct Analysis:** The 'M B' is redundant (already at B).
     The path is continuous: 'M A L B L C L D L A'. It is closed (ends
     at A) and has unique vertices A, B, C, D (4 vertices ->
     quadrilateral).

* **Error:** Counting the closing point as an extra vertex.
   * Path: 'M A L B L C Z' (a triangle).
   * Correct vertices: A, B, C (3 vertices -> triangle).
   * Incorrect count: A, B, C, A (4 vertices -> wrong shape).

**Options:** circle, heptagon, hexagon, kite, line, octagon, pentagon,
   rectangle, sector, triangle

**Now, analyze this path:**
<path d="[PATH_DATA]"/>
```

This optimized prompt achieves an accuracy of **96%**, demonstrating outstanding performance and highlighting the effectiveness of structured prompt optimization in reasoning-intensive tasks. The strength of this prompt lies in its explicit decomposition of the shape recognition process into interpretable reasoning steps. By first requiring path preprocessing (e.g., handling redundant 'M' commands) and then systematically parsing drawing commands, the model is guided to focus on structural consistency rather than surface-level token patterns. The careful instructions about avoiding duplicate vertices and distinguishing between closed and open paths reduce common sources of error, while the inclusion of concrete error–correction examples further grounds the model's reasoning. Finally, the restricted answer space with explicit options helps anchor the output format and prevents off-task generations. Together, these elements transform a vague classification task into a structured reasoning pipeline, leading to the significant performance improvement observed.

## A.5 API CALL COST ANALYSIS

We provide a theoretical estimate of the API call cost of our framework. The process involves sample pool initialization, node expansion, evaluation, analysis, rewriting, and rollout. We explicitly distinguish between evaluation calls and optimization calls.

**Notation.** Let

- $N_{\text{init}}$ be the number of API calls required to initialize the sample pool (evaluation model).
- $I$ be the number of MCTS iterations.
- $B$ be the number of nodes expanded per iteration (branching factor).
- $S$ be the number of samples evaluated per node in the testing stage.
- $R$ be the number of rollout steps per node.

We denote the unit cost of the evaluation model as $c_{\text{eval}}$ and the unit cost of the optimization model (analysis + rewriting) as $c_{\text{opt}}$.

**Node-Level API Calls.**   Each expanded node executes three stages:

1. **Testing:** $S$ evaluation calls
2. **Analysis:** 1 optimization call
3. **Rewriting:** 1 optimization call

Thus, the total API calls per node (excluding rollout) are

$$C_{\text{node}}^{\text{base}} = S + 2$$

with total cost

$$\text{Cost}_{\text{node}}^{\text{base}} = S \cdot c_{\text{eval}} + 2 \cdot c_{\text{opt}}.$$

**Rollout API Calls.**   During rollout, each node repeats the three stages for $R$ steps. Therefore, the total calls and cost per node including rollout are

$$C_{\text{node}}^{\text{total}} = (R+1) \cdot (S+2),$$

$$\text{Cost}_{\text{node}}^{\text{total}} = (R+1) \cdot \left( S \cdot c_{\text{eval}} + 2 \cdot c_{\text{opt}} \right).$$

**MCTS-Level API Calls.**   Across all nodes in all iterations:

$$C_{\text{MCTS}} = I \cdot B \cdot (R+1) \cdot (S+2),$$

$$\text{Cost}_{\text{MCTS}} = I \cdot B \cdot (R+1) \cdot \left( S \cdot c_{\text{eval}} + 2 \cdot c_{\text{opt}} \right).$$

**Total API Calls and Cost.**   Including sample pool initialization:

$$C_{\text{total}} = N_{\text{init}} + I \cdot B \cdot (R+1) \cdot (S+2),$$

$$\text{TotalCost} = N_{\text{init}} \cdot c_{\text{eval}} + I \cdot B \cdot (R+1) \cdot \left( S \cdot c_{\text{eval}} + 2 \cdot c_{\text{opt}} \right).$$

**Discussion.**   This formulation separates evaluation and optimization model costs, providing a parameterized estimate of computational requirements. The total cost grows linearly with the number of MCTS iterations $I$, the branching factor $B$, the rollout depth $R$, and the number of samples per node $S$, while the sample pool initialization $N_{\text{init}}$ contributes a fixed overhead.

