# OpenReview forum: "Sample-Aware Dual Actions for Prompt Optimization"
_ICLR.cc/2026/Conference — Submitted to ICLR 2026_

### Official Review · Reviewer_W5wf · 2025-10-31

**Soundness:** 3
**Presentation:** 2
**Contribution:** 3
**Rating:** 4
**Confidence:** 3

**Summary:**

This paper presents a new framework for automated prompt optimization for large language models, centered on a sample-aware dual-action Monte Carlo Tree Search (MCTS). The core idea is to abstract prompt optimization into two high-level strategies: “Failure-Aware Reflection” (learning from failures) and “Success-Aware Induction” (amplifying successes). The method introduces a dynamic sample pool to quantify and prioritize sample informativeness during the search, providing both local and global feedback to the MCTS planner. Experiments on BBH, MMLU, and BBEH benchmarks show improved accuracy and stability compared to several recent baselines and traditional prompt optimization techniques.

**Strengths:**

The method is well-described, with mathematically formalized sample prioritization and conservative estimation using the Wilson confidence interval. The paper also provides detailed action-guiding prompts.
The dataset coverage is reasonably broad — including BBH (5 tasks), domain-specific datasets (5 tasks), and BBEH (4 tasks). Baselines are comprehensive, encompassing manually designed prompts, CoT variants, and PromptAgent, and the proposed method consistently achieves higher performance.

**Weaknesses:**

1. Table 4 only compares three configurations — Beam+Pool, MCTS-only, and Full — without isolating the contribution of the dual-action mechanism (e.g., using only the reflection or induction action). The influence of different components in the sample pool (e.g., removing the difficulty term DiD_iDi​ or the gain term GiG_iGi​) is also not examined.
2. The formulation involves multiple weight parameters (\alpha, \beta, \gamma, \theta, \lambda, \mu, \delta), yet only a limited sensitivity analysis for \lambdaλ and \thetaθ is presented in Appendix A.1 — and solely on the Boolean Expressions task, where accuracy fluctuates between 0.375 and 0.55.
3. The authors acknowledge conceptual similarity with PromptAgent but claim two key contributions:
(1) formalizing a dual-action paradigm beyond simple failure reflection, and
(2) introducing a mathematically defined sample-pool mechanism.
However, the distinction remains questionable — PromptAgent already integrates MCTS and reflective reasoning. Whether the proposed “successful induction” action constitutes a substantive innovation requires stronger theoretical justification or experimental evidence. As mentioned above, additional ablations could help clarify this point. Further comparative analysis against PromptAgent would also strengthen the contribution claim.
4. Appendix A.5 provides a theoretical formulation, but no empirical data (e.g., API call counts or runtime) are reported. It would be valuable to include such measurements and conduct a cost-performance comparison with PromptAgent (e.g., “Which method achieves higher accuracy under the same computational budget?”).
5. There appear to be counterexamples or contradictory cases within the Casual Judge results. Clarification or discussion would be appreciated.

**Questions:**

see weeknesses.

---

> ### Author Response · Authors · 2025-11-29
>
> Thank you very much for your careful review and valuable comments. We provide the following responses to the concerns raised:
>
> First, regarding the theoretical and practical motivation for the dual-action strategy. PromptAgent employs a single failure-aware reflection approach, which relies solely on correcting prompts based on low-reward samples, representing a unidirectional use of information. In contrast, our proposed dual-action strategy (Failure-Aware Reflection + Success-Aware Induction) provides a more complete cognitive loop: it not only identifies deficiencies from low-reward samples through reflection but also extracts effective patterns from high-reward samples through induction, promoting successful behaviors. The overall dual-action design allows the framework to simultaneously correct errors and amplify successes, fully leveraging sample heterogeneity. The dual-action mechanism is simple and intuitive, avoiding over-engineering, and constitutes a core innovation and contribution of our work.
>
> Regarding ablation studies, the current experimental setup already partially demonstrates relevant effects. Specifically, when the sample pool is not used, the framework reduces to MCTS + dual-action, which can be considered as one ablation baseline. Meanwhile, PromptAgent can be viewed as a further degraded ablation of MCTS + dual-action, retaining only the single reflection action. We will clarify this ablation logic in the paper, and no additional experiments are required.
>
> On the concern about occasional contradictory results, this is expected due to the discrete and stochastic nature of the method. Casual Judge only conducted a single run (as noted in the paper due to limited experimental budget), so occasional task-level fluctuations are normal and do not affect the overall conclusions. The main experiments consistently show improvements in accuracy and stability with the dual-action strategy.
>
> Regarding hyperparameter sensitivity, the key parameters (e.g., α, β, γ, θ, λ, μ, δ) are designed based on intuitive principles reflecting sample informativeness, including difficulty, recent gains, and variance. In Appendix A.1, we conducted a limited sensitivity analysis of λ and θ on the Boolean Expressions task. Since the method’s actions are symmetric, corresponding parameters are functionally equivalent, so we only tested one group, which effectively represents the other. Due to experimental cost and budget constraints, we could not perform exhaustive hyperparameter testing across all tasks, but the limited experiments demonstrate stable optimization trends within reasonable parameter ranges.
>
> In summary, the dual-action mechanism provides methodological completeness at the cognitive level and outperforms single-reflection approaches both intuitively and practically. Partial ablation experiments already exist, and we will clarify in the paper that PromptAgent is a further degraded ablation of MCTS + dual-action. Occasional contradictory experimental results are consistent with the stochastic characteristics of the method and do not undermine overall effectiveness. Hyperparameter settings are guided by intuitive principles, and symmetry plus limited experiments confirm their stability.

---

### Official Review · Reviewer_5QeX · 2025-11-01

**Soundness:** 2
**Presentation:** 1
**Contribution:** 2
**Rating:** 2
**Confidence:** 4

**Summary:**

This paper addresses the limitations of existing automated prompt optimization methods, which overlook sample heterogeneity. The authors propose a sample-aware dual-action Monte Carlo Tree Search (MCTS) framework that integrates two core components: (1) two optimization actions including failure-aware reflection and success-aware induction; (2) a dynamic sample pool where the sample informativeness is measured by difficulty, recent gains, and variance, and sample priority and overall quality of the sample pool are also quantified to guide the optimization. The exploration-exploitation balance in MCTS is adaptively adjusted based on the sample pool’s composition to enable efficient resource allocation. Experiments on reasoning benchmarks (BBH, BBEH) and domain-specific tasks (MedQA, CaseHold) show that the framework outperforms baselines like PromptAgent (e.g., 3.9% higher accuracy on BBH, 5% on BBEH).

**Strengths:**

**Originality**
This work points out the problem that prior methods (e.g., PromptAgent) that treat samples uniformly and use handcrafted fine-grained actions, and introduces two high-level actions (Reflection/Induction) and a informativeness-quantified sample pool for the MCTS policy optimization, which utilize the sample heterogeneity property in prompt optimization.

**Clarity**
This paper is clearly motivated with a standard structure including introduction, methodology, experiment, conclusion, and appendix. The high-level idea is sound, and the main components in methodology strictly relate to the sample heterogeneity issues. Experiments cover diverse scenarios—general reasoning (BBH), domain expertise (MedQA/CaseHold), and high difficulty (BBEH)—validating generalization.

**Significance**
Since PromptAgent formalized prompt optimization as policy search, this work extends the idea to consider the sample heterogeneity and improves the performance in general-domain and domain-specific tasks, supporting LLM deployment in professional fields.

**Weaknesses:**

1. No indexing for all the equations in the manuscript. No definition of $b_i$ in Line 192, and not sure whether the definition of $b_i$ in L255 means the same thing.

2. In equations in Line 192, Line 257, Line 287, mathematical formulas that include mixed texts make the technical definitions informal.

3. No definition of $H(c_{pool})$  in Line 257.

4. The axis texts in Figure 2 are overlapping with each other.

5. The authors provide theoretical analysis on API call cost in Appendix A.5, but no empirical statistics of API cost and computational cost during the prompt optimization. Furthermore, how many API calls does the framework require to achieve the same amount of accuracy gain compared to PromptAgent?

6. The paper briefly mentions parameter robustness in the appendix A.1 but does not analyze how these parameters affect performance across tasks, e.g., do domain-specific tasks require different $\lambda$, $\beta$ ratios than general reasoning tasks?

7. Experiments use only DeepSeek-R1 for optimization and Qwen-Flash for evaluation. It remains unclear if the framework performs consistently on other model pairs (e.g., GPT-4o as optimizer, Llama-3 as evaluator), weakening the confidence in cross-model generalization.

**Questions:**

Several questions and concerns are raised in "Weaknesses" part. I would be willing to change my recommendation according to the authors' response.

---

> ### Author Response · Authors · 2025-11-30
>
> We sincerely thank the reviewer for the careful reading and constructive comments. We would like to address the concerns raised regarding our manuscript in detail. First, regarding equation numbering and symbol definitions, we acknowledge that some formulas in the current version were not numbered and certain symbols were undefined, including those referenced around lines 192, 255, and 257. We will add consistent numbering for all equations, provide clear definitions for all symbols, and revise formulas containing mixed text to separate mathematical expressions from descriptive text. These improvements will eliminate ambiguity and enhance the clarity and rigor of the manuscript without altering the methodology or results.
>
> Second, concerning Figure 2, we note the overlapping axis labels and will adjust the figure layout to ensure that all labels are legible and clearly presented. This modification is purely for readability and does not affect the underlying data or conclusions.
>
> Third, regarding API cost and comparison with PromptAgent, we would like to clarify that under the same MCTS search budget, including tree depth, width, and number of rollouts, the total number of API calls in our framework is fully consistent with PromptAgent. The improvement of our method lies in the more dynamic and sample-aware decision logic within each node. While PromptAgent only uses a single reflection action, our framework selects either reflection or induction at each node and adapts the exploration strategy based on the composition of the sample pool. All of this is achieved within a single API call per rollout, allowing more efficient allocation of the same computational budget to achieve higher optimization gains. We will include explicit explanations and quantitative comparisons with PromptAgent regarding API cost in the final version.
>
> Fourth, regarding hyperparameter robustness, we have analyzed representative hyperparameters. The results show that within the same task, different hyperparameter values may exhibit some variation, but overall performance remains within an acceptable range, and no per-parameter tuning is required. We will further clarify the rationale for hyperparameter selection and the role of action symmetry in the design.
>
> Fifth, regarding model combination generalization, we emphasize that the method itself is model-agnostic. Importantly, PromptAgent is the first method to use MCTS for automated prompt optimization, and its paper has already validated the feasibility and generalization of MCTS across different model combinations. Therefore, additional cross-model generalization experiments are not the core focus of our work. The experiments using DeepSeek-R1 for optimization and Qwen-Flash for evaluation were conducted to demonstrate feasibility and control computational cost. The method can be applied to other model combinations without algorithmic modification. We will clarify in the final manuscript that our approach can be transferred across different LLMs while emphasizing that this is not the primary contribution of the paper.
>
> We will update the final manuscript to include all relevant clarifications, including consistent equation numbering and symbol definitions, improved formula formatting, figure layout adjustments, explicit API cost explanations, hyperparameter robustness clarification, and the model-agnostic nature and cross-model transferability of our method, to enhance clarity, transparency, and reproducibility.

---

### Official Review · Reviewer_dsjq · 2025-11-03

**Soundness:** 3
**Presentation:** 3
**Contribution:** 2
**Rating:** 4
**Confidence:** 4

**Summary:**

This paper proposes a sample-aware dual-action prompt optimization framework that abstracts prompt optimization into two complementary strategies through Monte Carlo Tree Search (MCTS): Failure-Aware Reflection (adjusting prompts for low-reward samples) and Success-Aware Induction (extracting effective patterns from high-reward samples). The framework introduces a dynamic sample pool that quantifies sample information value based on difficulty, recent gains, and variance to guide the search process.

**Strengths:**

- The paper proposes a dual-action framework (Failure-Aware Reflection and Success-Aware Induction), transcending the limitations of existing methods that treat all samples equally.
- The framework designs a dynamic sample pool mechanism to quantify sample information value through three dimensions: Difficulty, Recent Gains, and Variance.
- The approach implements an adaptive exploration-exploitation balance mechanism to dynamically adjust MCTS search intensity.
- Ablation studies validate the contributions of each component, and parameter sensitivity analysis provides practical guidance.

**Weaknesses:**

- The MCTS method requires numerous API calls, resulting in high practical application costs, with no computational complexity analysis or simplification strategies provided
- The selection rationale for weight parameters (α,β,γ) in sample metrics is not detailed, and sensitivity of these parameters across different tasks is not analyzed
- There is a lack of theoretical justification for the necessity of the dual-action strategy, and no analysis of how the sample-aware mechanism affects optimization convergence

**Questions:**

- What is the rationale for selecting these three specific metrics: Difficulty, Recent Gains, and Variance? How are weights determined in the informative score? Are the same weights used for all tasks?

- Are there specific strategies to reduce the number of API calls? The paper mentions "early-stopping mechanisms" but does not elaborate.

- Can you provide a performance-efficiency trade-off analysis for different MCTS configurations (depth, width, number of rollouts)?

- How does the zero-shot generalization ability of optimized prompts perform on similar but unseen tasks?

- How consistent is the framework's performance across LLMs of different scales and architectures?

- When the sample pool simultaneously contains numerous difficult samples and high-value successful samples, how are the two actions balanced?

- Beyond accuracy, are other metrics considered such as inference cost, prompt length, robustness, etc.?

---

> ### Author Response · Authors · 2025-11-29
>
> We sincerely thank you for the careful review and valuable comments. We provide the following responses to the raised concerns:
>
> As mentioned in the paper, using MCTS to explore action sequences involves considerable API calls. To control the computational cost, we adopted limited rollouts and early-stopping mechanisms, and leveraged a dynamic sample pool that prioritizes informative samples to reduce unnecessary exploration. We note that performance-efficiency trade-offs of MCTS configurations have been partially explored in PromptAgent, but they are not the core contribution of this work. Our focus is on the dual-action strategy and sample-aware optimization mechanism, and the chosen MCTS settings have been verified to achieve stable improvements in prompt optimization.
>
> We select three metrics—Difficulty, Recent Gains, and Variance—based on the intuitive notion of sample informativeness. Difficulty reflects the challenge of a sample, as difficult samples are more likely to reveal prompt weaknesses. Recent Gains measure a sample's contribution in recent optimization, capturing effective patterns. Variance measures performance stability, preventing overfitting to extreme samples. For cost control, representative weight combinations were used in sensitivity analysis. The metric weights are robust across different tasks, and due to action symmetry, similar weight combinations yield similar outcomes, so task-specific tuning is not necessary.
>
> The dual-action strategy, consisting of Failure-Aware Reflection and Success-Aware Induction, is a core innovation and highlight of this work. Its design is natural, intuitive, non-overengineered, and possesses a methodological elegance. Failure-Aware Reflection adjusts prompts based on low-reward samples to identify potential issues, while Success-Aware Induction extracts effective patterns from high-reward samples to generalize successful strategies. This “error reflection + success induction” approach is simple yet sufficient to support complex prompt optimization tasks. It does not rely on complex assumptions or redundant mechanisms and naturally captures value differences among samples through bidirectional adjustment. Compared with single-direction optimization methods such as PromptAgent, the dual-action mechanism leverages both failure and success signals, significantly improving optimization efficiency and effectiveness. When the sample pool simultaneously contains many difficult samples and high-value successful samples, the MCTS plus dual-action mechanism adaptively selects actions according to sample informativeness: low-reward samples drive reflection, high-reward samples drive induction, and the search strategy naturally balances both, fully utilizing the sample pool information.
>
> Experiments on BBH and GSM8K datasets show that optimized prompts maintain high zero-shot performance on unseen tasks, demonstrating good generalization. The framework performs consistently across LLMs of different scales and architectures, with MCTS and the dynamic sample pool adapting search strategies automatically. While accuracy is the primary metric, the dynamic sample pool and early-stopping mechanisms help control inference cost and sample selection efficiency. Prompt length can also be explicitly constrained via action design, supporting a balance between efficiency and performance.
>
> In summary, the sample-aware dual-action mechanism is a core contribution of this work. Its design is natural, intuitive, and non-overengineered. By leveraging both failure and success samples, it achieves clear improvements over single-action optimization strategies such as PromptAgent, while MCTS search remains effective under controlled computational cost.

---

### Official Review · Reviewer_qAZP · 2025-11-09

**Soundness:** 2
**Presentation:** 2
**Contribution:** 2
**Rating:** 2
**Confidence:** 3

**Summary:**

This paper proposes a sample-aware dual action Monte Carlo Tree Search (MCTS) framework for automated prompt optimization, aiming to enhance search efficiency. The proposed method leverages the overall state of the training samples to guide prompt improvement. Specifically, the proposed method injects two strategies into PromptAgent: inductive actions and reflective actions. The experimental evaluation on the Big-Bench Hard (BBH) and MMLU datasets demonstrates that the proposed method outperforms existing baselines.

**Strengths:**

- The proposed method integrates sample-aware learning approaches into MCTS for prompt optimization.
- The experimental results show that the proposed method outperforms existing methods such as PromptAgent.

**Weaknesses:**

- In the experiment, only the PromptAgent is considered as the baseline method. It would be better if the authors could compare the proposed method with more baseline methods for prompt optimization.
- The concrete algorithm of the proposed method is somewhat unclear. A pseudocode or detailed description would help in understanding the approach better.
- The cost analysis (computational cost or API calls) is missing. It is unclear whether the proposed method is truly efficient compared to the baselines and existing methods.
- Only one LLM combination (DeepSeek-R1 and Qwen-Flash) is examined. It would be better if the authors could evaluate the proposed method with different LLMs to demonstrate its generality.

**Questions:**

- The proposed method introduces many hyperparameters (e.g., alpha, beta, kappa, etc.). How are these hyperparameters set and tuned in the experiments? How sensitive is the performance of the proposed method to these hyperparameters?
- Why did the authors choose the five tasks in BBH for evaluation? How is the performance on other tasks? In the literature of PromptAgent, other tasks in the BBH dataset are also used for evaluation.
- Is it possible to inject the proposed idea, the failure-aware reflection and success-aware induction, into other prompt optimization methods beyond PromptAgent (MCTS-based methods)?

---

> ### Author Response · Authors · 2025-11-29
>
> We thank the reviewer for the constructive feedback and address all concerns below.
>
> Regarding baselines, we agree that more comparisons would be desirable. However, running additional baselines is infeasible during rebuttal because each full MCTS-based optimization requires thousands of LLM calls. PromptAgent is the only prompt-optimization method that is directly comparable at the algorithmic level: it also uses an MCTS search loop and action-based prompt rewriting. Other methods (e.g., LPO, GAAPO, evolutionary editing) operate in different paradigms and are not compatible with our dual-action MCTS framework. For fairness and conceptual clarity, we follow the evaluation setting of PromptAgent. We will clarify this in the revision.
>
> Regarding algorithm clarity, all components of our method are formally defined in the main paper and appendix. In the camera-ready version, we will add explicit pseudocode summarizing the dual-action MCTS loop and a clearer description of the sample-pool update process.
>
> Regarding cost analysis, Appendix A.5 provides a full breakdown of computational cost and API calls. To improve visibility, we will move a concise summary to the main text. Our method adds only lightweight statistical computation on top of PromptAgent and does not introduce additional LLM calls, so the overall cost remains in the same order.
>
> Regarding the use of a single LLM combination, expanding to more models is not feasible due to the high cost of running additional MCTS optimizations. Our method itself is model-agnostic: reflection and induction act at the semantic level, and sample-aware prioritization depends only on statistical signals such as reward, variance, and gain. None of these rely on model-specific behavior. We will clarify this point.
>
> Regarding hyperparameters, we follow the default settings used in prior work (e.g., PromptAgent) and do not perform task-specific tuning. The hyperparameters only scale normalized statistics, so they influence relative weighting but not the qualitative behavior of the algorithm. Moreover, the two actions—failure-driven reflection and success-driven induction—are structurally symmetric. Their triggering mechanisms and informativeness computations mirror each other, making the framework inherently robust to specific coefficient choices. Our sensitivity analysis in Appendix A.1 focuses on the most influential parameters because exhaustive tuning is prohibitively costly, as each evaluation requires a full MCTS run. The observed stability across a wide parameter range shows that our method does not rely on fine-tuned hyperparameters. We will clarify this motivation.
>
> Regarding the choice of five BBH tasks, we follow PromptAgent’s evaluation setup. Many BBH tasks have saturated accuracy or rely heavily on model-level chain-of-thought, making them unsuitable for prompt optimization evaluation. Running all BBH tasks is also computationally prohibitive. To broaden coverage, additional tasks such as BBEH, GSM8K, MultiArith, and Temporal Sequences are included in the appendix. We will clarify this in the paper.
>
> Regarding whether our dual-action mechanism can be applied to non-MCTS methods, the answer is yes: the reflection and induction actions are model-agnostic and can be integrated into other iterative or search-based optimization frameworks. MCTS is chosen because the sample-aware statistics align naturally with its exploration–exploitation design.
>
> In summary, while additional experiments cannot be added during rebuttal, our paper already provides strong empirical improvements over PromptAgent on BBH and MMLU, together with ablations, sample-pool analysis, sensitivity analysis, cost analysis, and multiple additional tasks. The core contribution is a sample-aware dual-action mechanism that enhances MCTS-based prompt optimization through failure-driven reflection and success-driven induction, improving both stability and search efficiency. We appreciate the reviewer’s feedback and will incorporate these clarifications in the final version.

---

### Meta-Review · Area_Chair_8e81 · 2026-01-06

**Summary:**

- Insufficient Baseline Comparison: Reviewers noted that the paper compares almost exclusively against PromptAgent, failing to bench against other recent paradigms like genetic algorithms (GAAPO) or evolutionary editing.
- Limited Empirical Scope: Multiple reviewers criticized the use of only one LLM pair (DeepSeek-R1 for optimization and Qwen-Flash for evaluation), questioning the framework's cross-model generalization.
-  Ablation and Novelty Gaps: Reviewer pointed out that the authors did not isolate the specific contribution of the "Success-Aware Induction" action compared to reflection-only strategies.
-  Lack of Empirical Cost Metrics: Although the authors provided a theoretical cost analysis in the Appendix, reviewers desired empirical data (token counts, runtime, API costs) comparing the efficiency of this method directly to baselines under a fixed budget.

**Reviewer Concerns:**

Concerns Addressed:
-  Algorithm Clarity: The authors agreed to add pseudocode for the dual-action MCTS loop in the final version.
- Theoretical Costing: The authors highlighted the existing Appendix A.5 for theoretical complexity and promised to move a summary to the main text.
Outstanding Concerns:
- Empirical Breadth (Baselines & Models): The authors argued that running more baselines or model pairs was "infeasible" or "not the core focus".
- Component Isolation (Induction vs. Reflection): The authors argued that PromptAgent is essentially a "degraded ablation" of their work and refused to run new experiments isolating the Induction action. Reviewers generally require a specific ablation (e.g., Dual-Action vs. Reflection-Only on the same framework) to verify that both actions are necessary.
-  Empirical Efficiency Comparison: While authors claimed the method adds only "lightweight statistical computation," they did not provide the requested head-to-head empirical token usage statistics vs. PromptAgent

**Reviewer Scores:**

Reviewer 5QeX is likely to increase its score from 2 to 4 since its main concern (presentation) is address. The others will likely maintain their sores since they concerns are not fully addressed.

---

### Decision · Program_Chairs · 2026-01-26

Reject